# Genome-Wide Identification and Analysis of the Hsp40/J-Protein Family Reveals Its Role in Soybean (*Glycine max*) Growth and Development

**DOI:** 10.3390/genes14061254

**Published:** 2023-06-12

**Authors:** Muhammad Khuram Razzaq, Reena Rani, Guangnan Xing, Yufei Xu, Ghulam Raza, Muqadas Aleem, Shahid Iqbal, Muhammad Arif, Zahid Mukhtar, Henry T. Nguyen, Rajeev K. Varshney, Kadambot H. M. Siddique, Junyi Gai

**Affiliations:** 1Soybean Research Institute, MARA National Center for Soybean Improvement, MARA Key Laboratory of Biology and Genetic Improvement of Soybean, National Key Laboratory for Crop Genetics and Germplasm Enhancement, Jiangsu Collaborative Innovation Center for Modern Crop Production, Nanjing Agricultural University, Nanjing 210095, China; khuram.uos@gmail.com (M.K.R.); xinggn@njau.edu.cn (G.X.); 2022201025@stu.njau.edu.cn (Y.X.); 2National Institute for Biotechnology and Genetic Engineering, Faisalabad 38000, Pakistan; reenamaqsood@yahoo.com (R.R.); graza4@gmail.com (G.R.); marif_nibge@yahoo.com (M.A.); zahidmukhtar@yahoo.com (Z.M.); 3Center for Advanced Studies in Agriculture and Food Security (CAS-AFS), University of Agriculture, Faisalabad 38040, Pakistan; muqadasaleem@gmail.com; 4Horticultural Science Department, North Florida Research and Education Center, University of Florida/IFAS, Quincy, FL 32351, USA; shahidiqbalpak@hotmail.com; 5Division of Plant Sciences and National Center for Soybean Biotechnology, University of Missouri-Columbia, Columbia, MO 65211, USA; nguyenhenry@missouri.edu; 6Centre for Crop & Food Innovation, State Agricultural Biotechnology Centre, Food Futures Institute, Murdoch University, Murdoch, WA 6150, Australia; rajeev.varshney@murdoch.edu.au; 7The UWA Institute of Agriculture, The University of Western Australia, Perth, WA 6001, Australia

**Keywords:** Hsp40/J-protein family, *Glycine max* (L.) Merr., J-protein characterization, growth period, seed development

## Abstract

The J-protein family comprises molecular chaperones involved in plant growth, development, and stress responses. Little is known about this gene family in soybean. Hence, we characterized J-protein genes in soybean, with the most highly expressed and responsive during flower and seed development. We also revealed their phylogeny, structure, motif analysis, chromosome location, and expression. Based on their evolutionary links, we divided the 111 potential soybean J-proteins into 12 main clades (I–XII). Gene-structure estimation revealed that each clade had an exon-intron structure resembling or comparable to others. Most soybean J-protein genes lacked introns in Clades I, III, and XII. Moreover, transcriptome data obtained from a publicly accessible soybean database and RT-qPCR were used to examine the differential expression of DnaJ genes in various soybean tissues and organs. The expression level of DnaJ genes indicated that, among 14 tissues, at least one tissue expressed the 91 soybean genes. The findings suggest that J-protein genes could be involved in the soybean growth period and offer a baseline for further functional research into J-proteins' role in soybean. One important application is the identification of J-proteins that are highly expressed and responsive during flower and seed development in soybean. These genes likely play crucial roles in these processes, and their identification can contribute to breeding programs to improve soybean yield and quality.

## 1. Introduction

Plant growth period traits, flowering and maturity are complex traits controlled by several genes. They are essential ecological traits linked to photoperiod response and crop yield [1,2]. Some gene families, such as AP2/EREBP, NAC, and WRKY, rigorously combine gene regulatory systems in plant development processes, including senescence, hormone signaling, metabolism, and pathogen defense, which are well-developed in plants [3,4,5]. Flowering genes comprise a complicated network that controls the shift from vegetative to reproductive growth in flowering plants [6,7]. Based on molecular weight, heat shock proteins (HSPs) are divided into seven types: HSP40 (J-proteins DnaJ), HSP60 (GroEL), HSP10 (GroES), HSP100 (ClpB/A/C), HSP90 (HtpG), and HSP70 (DnaK). Additionally, J-domain protein J3 in *Arabidopsis* plays a role in integrating flowering signals [8]. *Arabidopsis* J-domain proteins are a broad and diverse class of molecular chaperones that depend on secondary structural assignments defined by invariable tripeptides and α-helices following the second helix. However, specific structural prerequisites and mechanisms of allostery of the functional interplay between DnaJ and Hsp70s remain limited. Alessandro Barducci presented findings on a conserved interaction surface created by the second helix of the J-domain and a connected region of Hsp70. This surface encompasses lobe IIA of the nucleotide binding domain, the linker between domains, and the β-sandwich of the substrate binding domain (SBD). The discovery was made through molecular simulations that utilized both coarse-grained and atomistic models, along with co-evolutionary sequence analysis. The validity of this interaction mode was corroborated by the latest X-ray structures, which depicted a complex formed between the J-domain of Escherichia coli DnaJ and its Hsp70 counterpart, DnaK [9,10]

HSPs have been researched in wheat, zebrafish, rice, and yeast [11,12,13,14]. HSP70, also known as DnaK, is a highly prevalent HSP in eukaryotic cells. It is crucial in protein quality control and relies on two co-chaperones to perform its functions. The initial co-chaperone is the nucleotide exchange factor (NEF), responsible for swapping the ADP molecule bound to HSP70 with ATP. This exchange event triggers the release of the folding substrate. Another co-chaperone, known as DnaJ, performs multiple functions. It identifies an unfolded substrate, transports it to DnaK, enhances ATP hydrolysis in HSP70s, and brings about stable structural alterations in chaperone proteins [15,16]. Hsp40s are universally present as co-chaperones of HSP70s in organisms. Their primary function is to uphold cellular protein equilibrium by actively participating in diverse cellular processes, including protein folding, assembly, stabilization, and translocation under normal circumstances. Moreover, they contribute to protein refolding and degradation when organisms encounter environmental challenges. Their fundamental roles have remained unchanged throughout evolution [17]. According to domain structure, J-proteins are categorized into four groups (Types A–D), but their evolutionary links are uncertain [17]. In *Arabidopsis*, almost 120 proteins have J-domains [18]. DnaJ serves as a disulfide isomerase protein to catalyze the creation of disulfide bonds and other isomers of disulfide bonds. Throughout evolution, the function of DnaJ has been considered conserved in organisms. 

J-domain proteins are engaged in biological activities localized in numerous subcellular plant parts [18]. Their biological function in plant growth and development is unclear, with few studies on J-proteins in plant development [19]. *Arabidopsis* DNAJ HOMOLOG 3 (*J3*), a transcriptional regulator, encodes a Type I J-domain in flowering. *J3* loss of function inhibits flowering due to reduced *FT* and *SOC1* expression. Type I J-domain proteins possess a modular sequence that consists of specific components: a J-domain, a domain rich in glycine and phenylalanine (G/F domain), a zinc finger domain known as CXXCXGXG, and a less conserved domain located at the C-terminal. In contrast, the remaining types of J-domain proteins are deficient in one or more of these domains. The arrangement of domains in type I J-domain proteins follows a sequential organization closely resembling the modular structure observed in DnaJ/Hsp40 [20]. J3 interacts with SVP in the nucleus, reducing *SVP*’s ability to bind to *FT* and *SOC1* regulatory sequences [8]. These findings suggest that *J3* detects flowering signals in various genomic paths and stimulates flowering [8]. Hsp40 plays multiple roles as an epigenetic regulator or enzyme and co-chaperones in various biological processes [21,22]. In *Arabidopsis thaliana*, DnaJ proteins are involved in plasma membrane H^+^-ATPase control, flowering signal integration, and mitochondrion-dependent plant growth [8,23]. To unveil the molecular mechanisms underlying plant growth and development regulated by the *EM-BRYONIC FLOWER (EMF)* 1 gene, a screening process was conducted to identify interacting proteins of *EMF1*. Using the yeast two-hybrid technique, a total of 11 proteins were detected. Among them, *EIP9, EIP6*, and *EIP1* were found to be three interacting proteins that encode proteins with DnaJ domain, B-box zinc-finger, and WNK kinase domains, respectively. Moreover, *EIP9*, *EIP6*, and *EIP1* mutants exhibited early flowering and greater expression during flower development and flowering time, while *EIP9*, *EIP6*, and *EIP1* overexpression transgenic plants exhibited late flowering. The results indicate that EMF interacts with *EIP9*, *EIP6*, and *EIP1* to control flowering time in *Arabidopsis* [24]. The DnaJ family is the most varied co-chaperone group, distinguished by conserved J-domains. These J-domains were initially discovered in *E. coli*’s DnaJ and are accompanied by a remarkably conserved HPD tripeptide signature motif [25,26].

DnaJ proteins consist of four distinct structural domains. The first domain, located at the N-terminus, is the J-domain. It is followed by a domain rich in Glycine and Phenylalanine (G-F), a Zn^2+^-finger domain, and a C-terminal domain that exhibits less conservation than the other domains. Approximately 70 amino acids form four α-helices within the J-domain region, namely helices I to IV. The J-domain possesses a distinctive structural characteristic of a highly conserved His-Pro-Asp (HPD) tripeptide motif located between helices II and III in the loop region. This motif plays a crucial role in facilitating the stimulation of HSP70 ATP hydrolysis. The interaction between the J-domain and HSP70 occurs specifically with the ATP-bound conformation of HSP70, precisely at the interface between the Nucleotide-Binding Domain (NBD) and the Substrate-Binding Domain (SBD). Once bound, the HPD motif establishes contact with crucial residues within the HSP70 ATP catalytic site. This interaction results in the remodeling of the NBD lobes, aligning the catalytic residues in an optimal position for ATP hydrolysis. Moreover, the J-domain also interacts with residues within the HSP70 SBD, leading to a high affinity for the HSP70 ADP-bound state and facilitating efficient capture of substrates. In summary, the J-domain’s involvement encompasses the activation of ATP hydrolysis through HPD motif interactions and promoting a strong affinity for the ADP-bound state of HSP70, ultimately enhancing substrate trapping. Furthermore, J-proteins can directly attach to substrates, promoting the selective interactions between HSP70 and specific polypeptides. As a result, this connection plays a crucial role in linking the functions of HSP70 to specific cellular processes [27]. 

Recent genomics developments (the availability of whole genome sequences and genetics and genomics databases) can detect DnaJ family genes in soybean, opening up new possibilities for investigating and confirming conserved and distinctive mechanisms of gene members during soybean growth. Preliminary studies into the HSP40 gene family have been carried out on the model plants rice and *Arabidopsis* [8,21], but limited information is available on this gene family in soybean. While the soybean genome is 1150 Mbp in size with 46,400 expected coding genes, the *A. thaliana* genome is 125 Mbp with 26,500 estimated coding genes. Thus, compared with *Arabidopsis*, the soybean genome is 9.2 times larger with 1.75 times more genes [28]. In the present study, we identified 111 J-protein coding gene transcripts and estimated their phylogenetic relationships, gene structure, motif analysis, domain analysis, chromosomal location, and expression profiling during soybean growth and development. We also provide the basis for functional studies on J-proteins in soybean. The findings of this study may serve as the foundation for a thorough examination of the DnaJ genes in soybean.

## 2. Materials and Methods 

### 2.1. Identification of J-Protein Members in Soybean

J-proteins in soybean (*Glycine max*) were detected using the plant genomics database Phytozome (https://phytozome-next.jgi.doe.gov/) (accessed on 8 August 2022). First, the DnaJ domain was blasted in the *G. max* genome (Wm82.a4.v1). The sequence files were filtered according to the following criteria: (1) select the longest transcript to denote each locus, (2) eliminate coding sequences < 150 bp, and (3) remove truncated and incomplete domain encoding genes [29]. After filtering, the sequences were submitted to a Hidden Markov Model (HMM) of a Pfam search of protein sequences to identify the protein domain family gene IDs [30], confirmed in the SMART database (http://smart.embl-heidelberg.de/) (accessed on 8 August 2022) and the National Center for Biotechnology Information (NCBI) Batch Web CD-Search Tool (https://www.ncbi.nlm.nih.gov/Structure/bwrpsb/bwrpsb.cgi) (accessed on 8 August 2022) [17].

### 2.2. Ancestral and Multiple Sequence Analysis

For the phylogenetic study, we used MEGA7 software [31] to align the full-length soybean J-protein gene amino acid sequences using default parameters and then created a neighbor-joining tree with 1000 bootstrap tests to divide the soybean J-proteins into diverse groups. 

### 2.3. Domain Association, Motif Analysis, and Gene Structure

The Gene Structure Display Server GSDS 2.0 (http://gsds.cbi.pku.edu.cn/index.php accessed on 10 August 2022) web tool identified the J-proteins’ exon-intron gene structure. The MEME tool detected ten conserved motif maximums in motifs ranging from 6 to 50 by uploading the protein sequences [32]. Subsequently, the protein sequences were analyzed using the NCBI conserved domain tool to identify and detect the domains associated with protein families [33].

### 2.4. Chromosomal Localization

The GFF3 General Feature Format file *G. max* Wm82.a4.v1 (accessed on 10 August 2022) containing accumulated genomic data for genes and other characteristics (RNA, DNA, and protein sequences) was downloaded from Phytozome. The TBtool (software) was used to create a map-like diagram of the DNA-J gene locations on all 20 chromosomes [34].

### 2.5. Patterns of Gene Expression

RNA sequence data for the estimated gene models in 14 tissues (flower, one cm pod, young_leaf, pod shell 10 DAF, pod shell 14 DAF, seed 10 DAF, seed 14 DAF, seed 21 DAF, seed 25 DAF, seed 28 DAF, seed 35 DAF, seed 42 DAF, nodule, and root) were downloaded from Soybase [35] to determine soybean J-protein expression. DnaJ gene expression data were used to develop a heat map. Further, RT-qPCR estimated the expression of DnaJ genes at different soybean developmental stages (leaf and fruit primordia and seed development stage). RNA isolation and purification were executed according to [36,37]. Actin (RP-II) was used as a reference gene to standardize the reaction [38]. RT-qPCR was performed according to the method described by [1,39]. Each qRT-PCR reaction (final volume, 20 μL) contained 8.6 μL ddH_2_O, 0.4 μL (100 nM) of forward and reversed primers, 10 μL SYBR Green II Master Mix, and 1 μL diluted cDNA. The reaction mixtures were heated to 95 °C for 4 min, followed by 35 cycles at 95 °C for 20 s, 60 °C for 20 s, and 72 °C for 40 s. For qRT-PCR, three biological replications were used, with the data calculated using the 2^−ΔΔCT^ method.

## 3. Results

### 3.1. Detection and Study of J-Proteins in Soybean

We detected 111 gene transcripts of J-proteins with conserved DnaJ domains after eliminating overlapped, truncated proteins, and non-targeted sequences, ranging from 162 to 1196 amino acids in length. The grand average of hydropathicity (GRAVY) ranged from –0.142 to –1.205. The molecular weight of proteins ranged from 17.47 to 86.14, with pI values ranging from 5.06 to 9.81, representing a shift from acidic to basic protein nature. Appendix A summarizes the gene information, including transcript ID, gene name, chromosome number, location on the chromosome, intron, and exon in the gene, amino acid length, molecular weight, pI value, GRAVY, and description. The phylogenetic analysis grouped the soybean J-proteins into 12 clades (I–XII) (Figure 1). Clades I, IV, IX, and XII had more than 10 members (multi-gene clades), while Clades II, III, VI, VII, VIII, and XI contained 7–9 members (oligo-genes). The remaining independent clades had 3–5 members (mono-genes). Figure 2 illustrates the phylogenetic analysis, gene structure, conserved motifs, and intron-exon distribution in the DnaJ gene family in *G. max*.

### 3.2. Multi-Gene Clades: Clades I, IV, IX, and XII

Clades I, IV, IX, and XII contained 14, 13, 13, and 12 genes, respectively, with few or no introns. Most Clade I members had the DnaJ domain and DUF 3444 (DUF: domain of the unknown function) (Figure 3). The Pfam HMM identified that all Clade IV members had a distinct DnaJ domain. Clade IV was divided into two subclades, with more introns in the genes in subclade IV-1 than in subclade IV-2. Furthermore, proteins of subclade II-1 frequently contained DnaJ-CXXCXGXG central domain embedded in the DnaJ-C domain N-terminus with motif CXXCXCXC linked to Zn^2+^. All Clade IX members had the distinct DnaJ-C-terminal domain, all with introns. All Clade XII members had the DnaJ domain. Clade XII was divided into two subclades, with more introns in subclade XII-1 than in subclade XII-2.

### 3.3. Oligo-gene clades: Clade II, III, VI, VII, VIII, XI

Clade II had seven members with DnaJ (DnaJ-C-terminal domain, DnaJ_CXXCXGXG), TPR_1, and zf-C2H2_jaz. All genes in this clade contained introns and exons. 

Clades III, VI, and VII include DnaJ, DUF, TPR, and J-domain proteins in the C-terminus, DnaJ_CXXCXGXG, DnaJ_zf, Fer4_9, Fer4_13, and Fer4_15 domains. Most Clade III proteins did not contain introns, but the clade had 31 exons. All genes in Clades VI and VII had introns and exons, but Clade VII *Glyma.10g158200* only had exons. Clade VI contained DnaJ_C terminal, DnaJ_CXXCXGXG, and DnaJ_zf superfamily domains. Molecular proteins of Clade III were present in the cytoplasm and nucleus and involved in protein folding, stress response, the transition from the vegetative to the reproductive stage, and heat shock protein binding. Proteins of Clade VI were present in the cytoplasm and mitochondria and involved in protein folding, sepal and petal formation, embryo sac development, synergid death, polar nucleus fusion, DNA-dependent DNA replication, and heat shock protein binding. Proteins of Clade VII were present in the mitochondria, chloroplast, and endoplasmic reticulum and involved in responses to high light intensity, heat, endoplasmic reticulum stress, heat shock protein binding, and hydrogen peroxide (according to gene ontology description). 

### 3.4. Unspecified Clades: Mono-Gene Clades V and X

The mono-gene clades had small dispersed branches with distant links, isolated above the well-classified clades. Clades V and X had 4–5 members, with introns and exons in gene structure. Clade V and X contained the DnaJ superfamily, Fer4_9, Fer4_13, Fer4_15, DnaJ_C, and DnaJ_zf superfamily domains. Clade V proteins were present in the chloroplast and involved in positive gravitropism, heat shock protein binding, and protein folding and unfolding. Clade X proteins were involved in protein folding and heat shock protein binding.

### 3.5. Chromosomal Position of J-Protein Genes in Soybean

The 111 soybean J-protein genes were mapped on all 20 chromosomes randomly dispersed within each clade (Figure 4). A maximum of nine genes (8.6%) was distributed on chromosome (chr) 8, with only two genes (1.9%) present on chr 17. No clade had genes on all chromosomes; Clade I had fourteen genes present on 11 chromosomes.

### 3.6. Soybean J-Protein Gene Expression Rates in Various Tissues

The expression of the soybean J-protein genes differed (Figure 5), with higher expression of genes *Glyma.03G218300*, *Glyma.03G232700*, *Glyma.07G150700*, *Glyma.08G109700*, *Glyma.09G075400*, *Glyma.10G094400*, *Glyma.11G241000*, *Glyma.11G077400*, *Glyma.15G183400*, *Glyma.15G149100*, *Glyma.17G078100*, *Glyma.19G215100*, *Glyma.19G180600*, and *Glyma.20G141300* in almost all studied organs and tissues, most of which had J-domain and other domains. Genes *Glyma.01G166000*, *Glyma.03G232700*, *Glyma.08G109700*, *Glyma.08G337100*, *Glyma.09G075400*, *Glyma.10G094400*, *Glyma.11G077400*, *Glyma.13G289600*, *Glyma.15G183400*, *Glyma.16G127700*, and *Glyma.17078100* had relatively greater expression in flowers, with 91 soybean genes expressed in at least one tissue. Furthermore, six genes were randomly selected to study their expression using RT-qPCR (Figure 6). *Glyma.15G183400* and *Glyma.07G150700* were highly expressed in leaf primordia. *Glyma.11G241000* and *Glyma.19G215100* were highly expressed in flower primordia. *Glyma.03G232700* and *Glyma.01G166000* were highly expressed in the seed development stage. Hence, the RT-qPCR results resembled the RNA sequence data.

## 4. Discussion

### 4.1. Role of DnaJ in Plant Growth and Development

J-domain proteins capture and transport proteins requiring folding or refolding to HSP70 for further processing. HSP70s do not function independently in the protein folding mechanism; instead, they collaborate with J-proteins to form a complex that acts as a folding machine. HSP40s have been categorized into four classes due to their resemblance to the domain structure of DnaJ [16,40]. *GmDNJ1* was extensively expressed under ideal growth circumstances and significantly activated in abiotic stress treatments. *GmDNJ1* (a type-I HSP40) is important for soybean plant development and heat tolerance, likely due to its role in monitoring misfolded proteins for refolding to maintain the range of cellular activities [41]. The *BIL* gene encodes a protein belonging to the mitochondrial Hsp40 family. When the BIL2-OX plants were subjected to a Brz treatment, which is known to suppress brassinosteroid plant hormones, the overexpression of BIL resulted in increased cell proliferation. This, in turn, promoted the growth of plant roots and inflorescences. Another study showed that BIL2 stimulates cell growth during BR signaling by promoting mitochondrial ATP production [42]. Furthermore, the expression and accumulation of heat shock proteins (HSPs) were observed in the flowers, pods, and seeds of *Medicago sativa* throughout a typical growing season. The authors noted that HSPs were frequently expressed in reproductive structures, suggesting their vital contribution to successful reproduction [43]. Rice leaf mutant 7 controls leaf size by coding for the heat shock protein OsHSP40. Rice plants with disrupted Oshsp40 genes exhibit significantly smaller leaves than wild-type plants [44].

### 4.2. Detected Soybean J-Proteins Differ from Those Reported in Other Crops 

Zhang et al. (2018) identified 129 J-proteins in *Brassica oleracea*, grouping them into 15 major clades (I–XV) using phylogenetic studies with domain organization and gene structure as references. The clades were divided into several undefined branches with distinct relationships. The J-proteins from diverse clades operated together or individually to form a complex regulatory network [17]. Another study identified 104 J-protein genes in rice [45]. 

After reviewing the published results, a study conducted in 2019 reassessed the rice J-protein members within the Phytozome database. During this process, they identified 11 previously unknown genes and gathered a total of 115 J-protein genes from rice [21]. The J-proteins identified were categorized into nine clades (I–IX) based on their phylogenetic relationships, exhibiting molecular weights from 10.20 to 287.69 kDa. Gene-structure investigations indicated that members within each clade possessed a similar or comparable exon-intron structure, with a notable observation that several rice J-protein genes in Clade VII lacked introns. Chromosome mapping studies further unveiled tandem duplication events occurring throughout evolution. According to the expression profile, at least one tissue expressed 61 rice J-protein genes, suggesting their involvement in rice growth and development. Following the identification of 96 differentially expressed genes using RNA-seq, 62.50% (60/96), 67.71% (65/96), and 59.38% (57/96) of the genes were stimulated by salt, drought, and heat stress [21]. 

Here, we identified 111 J-proteins in soybean, which were grouped into 12 clades based on phylogenetic relationships and belonging to oligo-gene, mono-gene, and multi-gene clades according to the number of members (Figure 1). The identified protein genes in soybean had molecular weights ranging from 17.31 to 127.33 kDa (Appendix A). Transcription data and RT-qPCR determined the differential expression of these genes in various tissues and organs (Figure 6). This work offers a useful perspective on J-protein classification and a starting point for additional functional studies of J-protein genes in soybean.

### 4.3. J-Proteins Are Involved in Their Associated Networks during Soybean Growth and Development

J-proteins that operate with the Hsp70 molecular system play a role in the regulatory networks governing plant growth, development, and responses to environmental challenges, such as abiotic stress. The DNAJ HOMOLOG 3 (*J3*) of *A. thaliana* functions as a major flowering promoter to suppress the translation of ft, a flowering locus, and repress the upregulation of constans (SOC1) via its connection with the short vegetative phase (SVP), facilitating flowering stimuli. As a result, J3 stimulates flowering by upregulating SOC1 and FT expression [8,46]. 

Research has shown that *A. thaliana* TMS1 generates a heat shock protein (Hsp) similar to AtERdj3A, a J-protein. This J-protein, AtERdj3A, plays a crucial role in developing pollen tubes and contributes to the thermotolerance of diverse plant tissues [47]. A study on soybean flowering identified multiple regulators engaged in a complex state of a floral shift against diverse developmental and environmental stimuli recognized by several flowering genetic pathways [48]. In *Arabidopsis*, the THERMOSENSITIVE MALE STERILE 1 protein interacts with BiPs through its DnaJ domain, enhancing their ATPase enzyme activities [47].

The study employed an in silico analysis approach to identify and conduct a genome-wide analysis of the DnaJ gene family in soybean. From wet lab experiments, we have included RT-qPCR analysis. However, we could not conduct a comprehensive functional study of these proteins due to technical constraints and limited available resources. Future research efforts should focus on developing alternative approaches or acquiring additional resources to address this limitation and gain deeper insights into the functional significance of the J-protein family in soybean.

## 5. Conclusions and Future Perspective

We identified 111 potential soybean J-proteins—divided into 12 main clades (I–XII) according to their evolutionary links—dispersed randomly on all 20 chromosomes. The gene-structure analyses revealed that the 12 clades contained introns. Furthermore, RNA-sequencing revealed 98 genes with differential expression, with 61.2, 58.2, 56.1, 58.2, 42.95, 40.8, 43.9, 40.8, 32.7, 51, 48.9, 46.9, and 29.6% expressed during flower, young leaf, one cm pod, pod shell 10DAF, pod shell 14DAF, seed 10 DAF, seed 14DAF, seed 21DAF, seed 25DAF, seed 28 DAF, seed 35 DAF, seed 42 DAF, root, and nodule development, respectively. These findings indicate that J-proteins play a role in soybean growth and development and offer a foundation for the functional study of J-proteins in soybean. 

The application of this study lies in the characterization and understanding of the J-protein gene family in soybean. By investigating the phylogeny, structure, motif analysis, chromosome location, and expression patterns of J-protein genes, results depicted the role of these genes in soybean growth and development. The division of the soybean J-proteins into 12 main clades based on their evolutionary links allows for a better understanding of their functional diversity. This information can guide future studies to investigate the specific roles of J-proteins within each clade and explore their potential applications in enhancing soybean traits. Characterizing the gene structure revealed similarities and differences among the clades, providing insights into the evolutionary patterns of J-protein genes in soybean. This knowledge can aid in identifying conserved functional domains and regulatory elements, which can be utilized in genetic engineering approaches to modulate J-protein activity and improve soybean performance. Examining the differential expression of DnaJ genes in various soybean tissues and organs provides a comprehensive overview of their spatial distribution and suggests their involvement in the soybean growth period and seed development. This information can be utilized to target specific tissues or developmental stages for further functional research and investigate the potential manipulation of J-proteins to optimize soybean growth and productivity.

Overall, this work offers a foundation for further functional research on J-proteins in soybean and provides valuable information for plant scientists, breeders, and genetic engineers interested in improving soybean traits related to growth, development, and stress responses. A significant application of this research is the identification of J-proteins that exhibit high expression and responsiveness during flower and seed development in soybean. These genes are believed to have vital functions in these processes, and their identification holds the potential for enhancing soybean breeding programs focused on increasing yield and improving overall quality.

## Figures and Tables

**Figure 1 genes-14-01254-f001:**
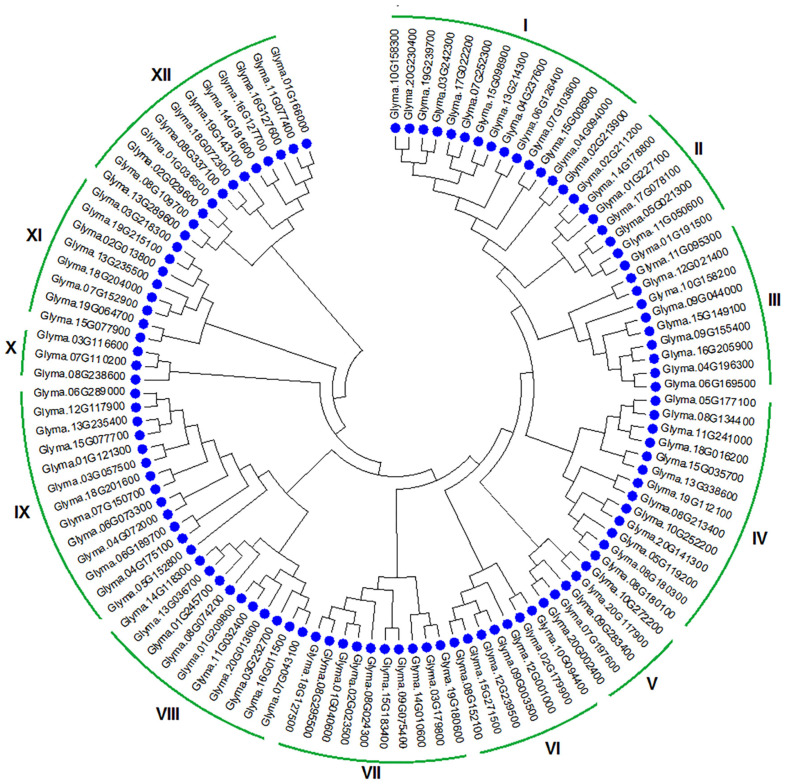
Phylogenetic study of soybean J-proteins. Clustal X was used to align 111 soybean J-proteins using full-length amino acid sequences, while MEGA6 software created the phylogenetic tree using the 1000 bootstrap neighbor-joining option. The J-proteins in soybean were grouped into 12 clades: multi-gene clades (I, IV, IX, and XII), oligo-gene clades (II, III, VI, VII, VIII, XI), and mono-gene clades (V and X).

**Figure 2 genes-14-01254-f002:**
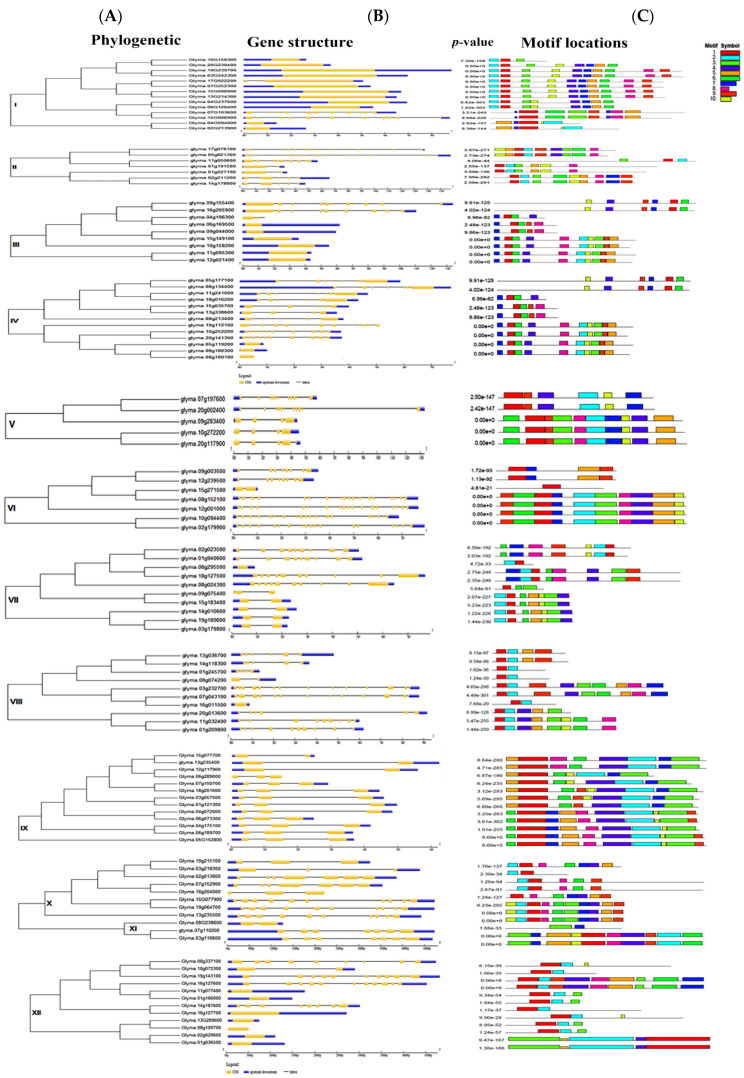
Phylogenetic analysis, gene structure, conserved motifs, and intron, exon, distribution in the DnaJ gene family in *G. max*. (**A**) Ancestral relationship of the DnaJ gene family categorized into 12 clades; (**B**) Exons (yellow boxes), untranslated region (UTR) introns (black lines), and upstream and downstream sequences (blue boxes); (**C**) Motif detection in DnaJ genes with *p*-value (a total of 10 motifs are mentioned with different colors).

**Figure 3 genes-14-01254-f003:**
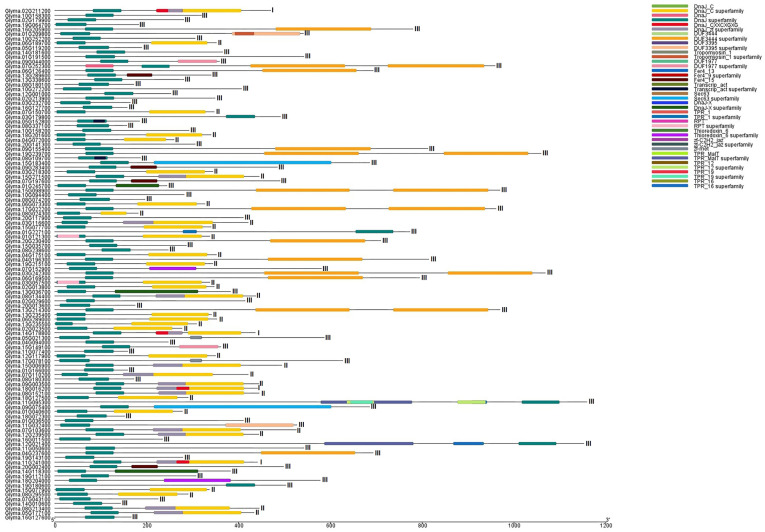
Domain organization of DnaJ genes in *G. max.* DnaJs are further categorized into three distinct types. I for type-I proteins primarily comprise three extensively conserved domains. The first domain is the N-terminal J-domain (DnaJ), which binds to the ATPase domain of HSP70. The second domain is the zinc-finger domain (CXXCXGXG), and the third domain is the C-terminal domain (DnaJ_C). On the other hand, II for type-II proteins lack the zinc-finger domain, while III for type-III proteins only possesses the J-domain. In type III proteins, the J-domain can be located at any position along the length of the protein, and there is a possibility of additional domains apart from the J-domain.

**Figure 4 genes-14-01254-f004:**
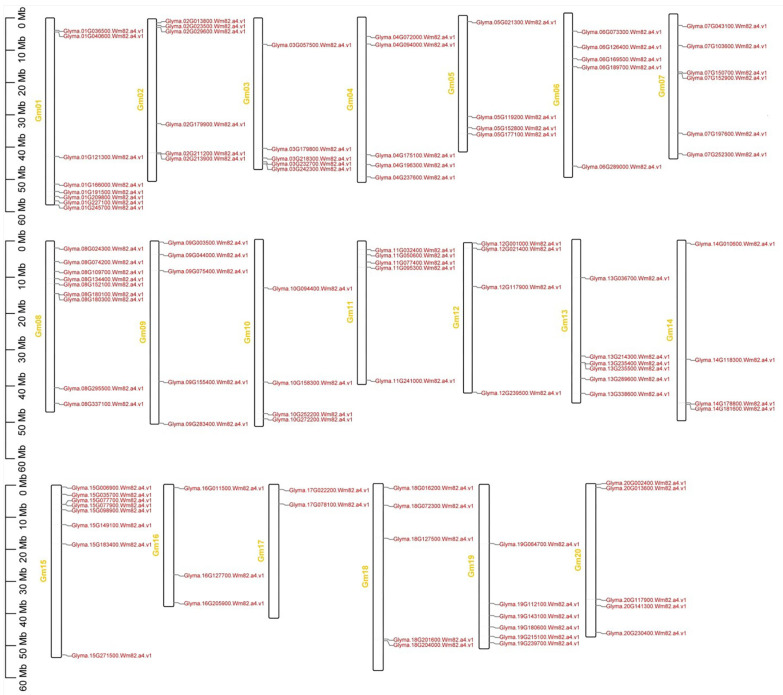
Location of DnaJ genes in the soybean genome. The chromosomal location of each gene was mapped to the soybean genome. Left side of each chromosome, scale represents gene position in megabases (Mb); right side, gene names correlate with probable placements of each DnaJ gene.

**Figure 5 genes-14-01254-f005:**
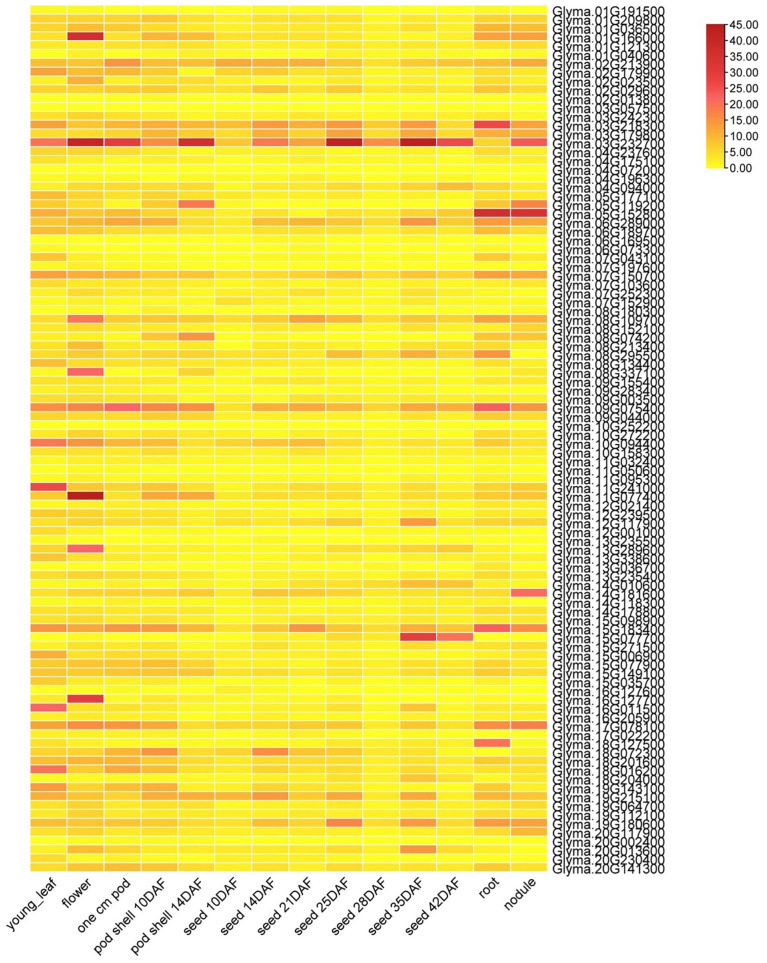
Soybean J-protein gene family expression levels in numerous tissues, organs, and developmental stages. RNA sequence data for each predicted gene model in 14 tissues (young leaf, one cm pod, pod shell after 10 days of flowering, pod shell after 14 days of flowering, seed after 10 days of flowering, seed after 14 days of flowering, seed after 21 days of flowering, seed after 25 days of flowering, seed after 28 days of flowering, seed after 35 days of flowering, seed after 42 days of flowering, nodule, and root) from https://soybase.org (accessed on 14 August 2022) were used for gene expression profiles. The scale of colors shows lower to higher FPKM values.

**Figure 6 genes-14-01254-f006:**
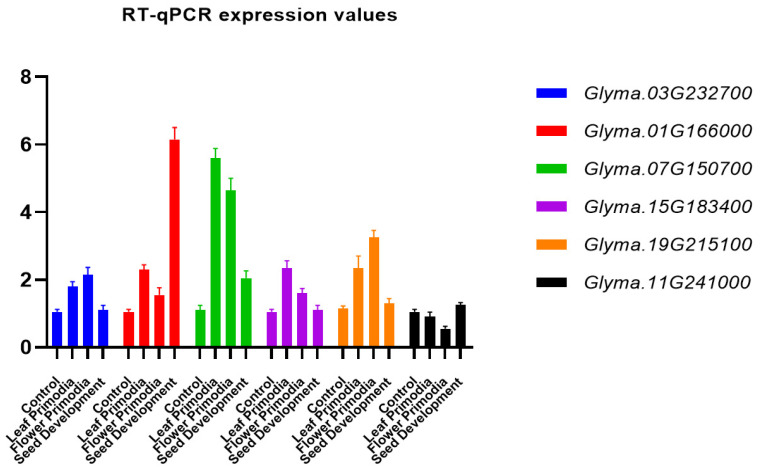
Expression of DnaJ genes was estimated through qRT-PCR. The DnaJ gene expression levels were set to 1 in control seedlings. The error bar indicates the standard error of their biological replicates.

## Data Availability

The article and Appendix A contain the data.

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
