# Peer review of "Genome-Wide Identification and Analysis of the Hsp40/J-Protein Family Reveals Its Role in Soybean (Glycine max) Growth and Development"

_genes, 2023, doi:10.3390/genes14061254_

Round 1
Reviewer 1 Report
The authors have performed in silico analyses. Such studies often need to be validated through wet laboratory experiments. I suggest functional characterization of 1-2 interesting genes (over-expression or mutant lines). This will add some value to paper.
I have few specific comments,
1. If an acronym is used in the abstract or title, it must be spelled out. DnaJ, hsp40, J protein … etc
2. The manuscript needs revision for language and grammar.
3. Abstract is not describing all key results/finding. Looks like study outline. I suggest re-writing. Include applications of the research.
4. What is applicability of this research? Which research question is addressed here? Include detailed information.
5. Why only 6 genes selected for RT-PCR?
6. The authors have performed in silico analyses. Such studies often need to be validated through wet laboratory experiments. I suggest functional characterization of 1-2 interesting genes (over-expression or mutant lines). This will add some value to paper. (Important)
7. Line 137. RNA sequence or sequencing?
8. Line 140. What is standardized data?
9. Figure 6. What these number denotes? 0-45
Extensive editing of English language required
Reviewer 2 Report
This manuscript deals with the characterization of hsp40/J-proteins encoded in the soybean (Glycine max) genome. The authors identified 111 J-proteins that were grouped into 12 clades (I–XII), according to sequence-based phylogeny analysis. Also, using publicly available RNA-seq data, the authors analyze changes in the expression of those hsp40 genes in different soybean tissues and organs. Also, the authors performed qPCR validation and some data are shown. At first glance, the manuscript contents seem superb; however, the poor elaboration of the manuscript and the figures, together with the deficiencies in the methodological descriptions preclude a favourable recommendation.
A list of comments and suggestions follows.
The INTRODUCTION section needs to be improved. Some points to consider:
1. Lines 47-48. Please, be more precise in indicating what J-protein(s) are you referring to (according to the cited article).
2. Line 50. Could the authors be more precise in describing the typical structural features of J-proteins?
3. Lines 52-53. It seems that authors have included in brackets the names receiving the mentioned proteins in Escherichia coli, but this fact is only deduced after reading "in Eschericihia coli IbpA/B". It would be more appropriate to indicate this when referring to the E. coli orthologue HSP40 (the first protein mentioned in the list).
4. Lines 57-58. According to Zhang et al., 2018, there are 117 J-proteins in Arabidopsis. Please, check and revise.
5. Lines 60-63. Are you referring to the E. coli DnaJ protein or to every J-protein? Please, be more precise.
6. Lines 67-68. Please, describe what a type I J-domain is.
7. Line 72. The sentence 'DnaJ—also known as J-proteins and Hsp40 family' is incorrect. As you mentioned DnaJ is the name used to refer to the first described bacterial Hsp40 and founder member of the Hsp40 family. It would be more appropriate to write: Proteins of the Hsp40 family are also known as J-proteins or DnaJ-domain proteins (JDP).
8. Line 85. What are the authors mean by the sentence: Recent genomics developments?
9. Line 93. Check: The current work uses (?)
10. Lines 94-95. Please, be more precise. What means: 'the corresponding J-proteins'?
11. Line 99. Please, be more precise. DnaJ is the name used for the bacterial Hsp40 protein (see point 7).
MATERIALS AND METHODS.
12. Section 2.2. (Protein identification and analysis) maybe deleted, as it does not contain any methodological information.
13. Section 2.6. The description of RNA-seq data used in this study should be extensively improved. For instance: do all samples derive from the same study?, how many replicates were used?, how data were normalized?, and so on.
14. Lines 129-130. Please rewrite the sentence: The protein sequences were then subjected to an NCBI conserved domain to detect the protein family domains
15. Lines 140-141. Check: data was
16. Line 141. Check: RT-qPCR estimated
RESULTS
17. According to structural features, HSP40s are classified into four types (types I–IV). Detailed information on this classification may be found in the following review articles (among others):
- Cheetham, M.E., and Caplan, A.J. (1998). Structure, function and evolution of DnaJ: conservation and adaptation of chaperone function. Cell Stress Chaperones 3, 28–36. PMID: 9585179
- Solana et al (2022). The Astonishing Large Family of HSP40/DnaJ Proteins Existing in Leishmania. Genes (Basel). 13, 742. PMID: 35627127.
In this regard, the authors should try to classify the 111 J-proteins into these four categories (I-IV). This analysis is relevant as the presence or absence of particular structural motifs in a given J-protein would provide clues about its functional role in the HSP70 activity cycle.
18. The legend to figure 2 lacks descriptions of the meaning of several features shown such as the p-value and the motifs identified (1 to 10). Moreover, in the legend, panels A, B and C are mentioned, but they are not shown in the figure.
19. Lines 185-186. Please, rephrase the sentence: four cysteine-rich repetitions of the motif CXXCXGXG, each associated with one unit of Zn2+
20. In the legend to figure 3, it should be indicated the repository in which the listed motifs may be found.
21. Lines 199-207. Appropriate references should be added.
22. Lines 239-240. The authors should show the qPCR results or alternatively to eliminate this sentence.
23. In legend to figure 5, the meaning of the scale of colors should be explained.
24. The DISCUSSION section should be extensively revised (and shortened).
Round 2
Reviewer 1 Report
The authors have addressed most of the comments raised by me,
1) Point 6 raised by has not been addressed.
“Point 6: The authors have performed in silico analyses. Such studies often need to be validated through wet laboratory experiments. I suggest functional characterization of 1-2 interesting genes (over-expression or mutant lines). This will add some value to paper. (Important)”
Response 6: The study employed an in-silico analysis approach to identify and conduct genome-wide analysis of the DnaJ gene family in soybean. We have already included RT-qPCR analysis. However, due to further practical constraints, the functional confirmation of these genes could not be carried out at this time.
Authors need to give this explanation in manuscript also.
2) Figure 5. What is the unit of gene expression here? TPM? RPM or TPM? Provide the information in legend.
3) Minor editing of English language required.
4) Check the resolution of figures. Fig 2,3 and 4
Minor editing of English language required.
Reviewer 2 Report
After reading the authors responses to my previous comments and the revised manuscript, I consider that the authors have not sufficiently addressed my suggestions. Thus, the manuscript remains suboptimal and I maintain the recommendation of rejecting it.
Among the extensive list of suggestions contained in my previous report, I will refer to those considered as compulsory, but authors have not addressed properly.
A) [authors’ response to point 15]. Section 2.6. The description of RNA-seq data used in this study should be extensively improved.
The authors’ response: The RNA-sequence data we have used in this study was obtained from the soybean genetics and genomics database (http://soybase.org) this data is freely available
This response is not valid. The authors should identify the study (or studies) IDs, the features of RNA-seq data, the number of replicates and so on.
B) [point 18]. The description of RT-qPCR assays remains quite incomplete. Moreover, controls are missed or not described.
C) [point 19]. In this regard, the authors should try to classify the 111 J-proteins into these four categories (I-IV). This analysis is relevant as the presence or absence of particular structural motifs in a given J-protein would provide clues about its functional role in the HSP70 activity cycle.
The authors’ response: Thank you for the improvement of the manuscript, we have checked these articles and also incorporated in the manuscript and cited.
This point was not raised with the objective of imposing a citation of the mentioned articles. The authors were asked to perform a structural analysis looking for the typical structural domains existing in J-proteins. This analysis has not been included in the revised manuscript.
There are other points that have not been addressed properly, but it is not a question of repeating them again.
Round 3
Reviewer 2 Report
-